# Effect of genetically predicted sclerostin on cardiovascular biomarkers, risk factors, and disease outcomes

Marta Alcalde-Herraiz[1,2], JunQing Xie [1], Danielle Newby[1], Clara Prats [2], Dipender Gill [3], María Gordillo-Marañón [4,5], Daniel Prieto-Alhambra [1,6] ✉, Martí Català[1,7] & Albert Prats-Uribe [1,7]

Sclerostin inhibitors protect against osteoporotic fractures, but their cardio-vascular safety remains unclear. We conducted a *cis*-Mendelian randomisation analysis to estimate the causal effect of sclerostin levels on cardiovascular risk factors. We meta-analysed three GWAS of sclerostin levels including 49,568 Europeans and selected 2 SNPs to be used as instruments. We included heel bone mineral density and hip fracture risk as positive control outcomes. Public GWAS and UK Biobank patient-level data were used for the study outcomes, which include cardiovascular events, risk factors, and biomarkers. Lower sclerostin levels were associated with higher bone mineral density and 85% reduction in hip fracture risk. However, genetically predicted lower sclerostin levels led to 25–85% excess coronary artery disease risk, 40% to 60% increased risk of type 2 diabetes, and worse cardiovascular biomarkers values, including higher triglycerides, and decreased HDL cholesterol levels. Results also suggest a potential (but borderline) association with increased risk of myocardial infarction. Our study provides genetic evidence of a causal relationship between reduced levels of sclerostin and improved bone health and fracture protection, but increased risk of cardiovascular events and risk factors.

Osteoporosis is one of the most common chronic conditions worldwide[1–3], resulting in a high risk of fractures, including hip fractures. Hip fractures, in turn, have a great impact on health-related quality of life, as well as an increase in morbi-mortality and healthcare costs[4,5]. Osteoporosis-related fractures can be prevented by treating the underlying bone fragility. International guidelines recommend anti-resorptives (e.g. bisphosphonates or denosumab) as first line therapies for most patients, with anabolic therapies (e.g. teriparatide or romoso-zumab) used to treat more severe cases[6]. However, several osteoporosis treatments have been previously associated with an increased risk of

cardiovascular adverse events[7]. For example, strontium ranelate was found to be associated with an increased risk of serious cardiovascular safety events and deaths[8] leading to the European Medicines Agency (EMA) recommending several new contraindications and restrictions of use[9]. Separately, Odanacatib, a newly developed cathepsin K inhibitor, was shown effective to improve bone density and reduce fracture risk but increased the risk of stroke in phase 3 trials leading to the discontinuation of the development of this therapy[10].

More recently, sclerostin inhibition has been clinically investigated as a potential therapeutic target for osteoporosis, and two

[1]Centre for Statistics in Medicine and NIHR Biomedical Research Centre Oxford, NDORMS, University of Oxford, Oxford, UK. [2]Computational Biology and Complex Systems (BIOCOM-SC), Department of Physics, Universitat Politècnica de Catalunya, Castelldefels, Spain. [3]Department of Epidemiology and Biostatistics, School of Public Health, St Mary's Hospital, Imperial College London, London, UK. [4]Institute of Cardiovascular Science, Faculty of Population Health, University College London, London, UK. [5]Data Analytics and Methods Task Force, European Medicines Agency, Amsterdam, Netherlands. [6]Department of Medical Informatics, Erasmus University Medical Centre, Rotterdam, the Netherlands. [7]These authors contributed equally: Martí Català, Albert Prats-Uribe. ✉e-mail: daniel.prietoalhambra@ndorms.ox.ac.uk

medicines (blosozumab and romosozumab) have shown to improve bone strength and reduce fracture risk[11,12]. After four phase 3 randomised controlled trials[13–17], romosozumab was approved for the treatment of severe osteoporosis in most parts of the world. While two placebo-controlled trials showed no evidence of cardiovascular safety concerns (FRAME[14], STRUCTURE[15]) a subsequent head-to-head trial comparing romosozumab to alendronate showed a potential increased risk of myocardial infarction and stroke in the group treated with romosozumab (ARCH[16]). This imbalance was also seen in a placebo-romosozumab trial in men (BRIDGE[17]). Considering this potential safety concern, the EMA imposed contraindications to minimise any possible risk and requested an observational international post-authorisation safety study[18].

Some observational studies have also reported similar findings. A meta-analysis of clinical trials and human genetics[19] indicated a potential higher risk of cardiovascular adverse events, and other risk factors such type 2 diabetes. These associations were also supported by a recent Mendelian randomisation study[20], where it was found that lower sclerostin levels were causally related with increased coronary artery disease and myocardial infarction risk. However, no evidence was found of these associations in another population-based phenome-wide association (PheWAS) study[21].

In the absence of additional clinical trials and follow-up post-authorisation safety studies, the use of genetic studies to evaluate the safety profiles of therapeutic targets is gaining popularity due to the growth of readily accessible data from genome-wide association studies (GWAS)[22]. Mendelian randomisation (MR) uses genetic variants that are associated with an exposure as instrumental variables to assess the causal influence of the exposure on an outcome[23]. Since genetic variants are randomly allocated at conception, this approach can avoid or minimise confounding by indication. Specifically, when the exposure under investigation is a drug target, the selection of instruments is usually restricted to those around the gene encoding the drug target to reduce the risk of horizontal pleiotropy[24]. MR can therefore be used to triangulate the evidence for a causal relationship between sclerostin inhibition on cardiovascular risk[25].

In this study we meta-analysed previously published genetic data to create the largest dataset of genetically predicted sclerostin levels. We aimed to conduct a two-sample *cis*-Mendelian randomisation study to estimate the causal effect of sclerostin levels on a variety of cardiovascular outcomes and investigate potential pathways. The novelty of this study relies on the use of the most up to date sclerostin GWAS (at the time of the study) for the IVs selection, and the inclusion of UK Biobank data for evidence triangulation. We also study effects on biomarkers, cardiovascular traits, and risk factors to provide insights about the underlying mechanistic insights. Moreover, we performed colocalization analyses to study the probability of a shared causal variant between the investigated traits.

Given the existing information on the genetic associations of sclerostin levels and all the outcomes of interest, this study builds on existing studies to generate evidence to inform clinicians, patients, and medicine regulators globally.

## Results
### Meta-analysis
There were 49,568 European individuals and a total of 551,580 SNPs from chromosome 17 included in the meta-analysis of sclerostin GWAS (Supplementary Fig. 1). We found little evidence of genomic inflation ($\lambda = 1.04$) in the meta-analysed fixed-effects dataset, suggesting that there was little residual genetic bias between the meta-analysed datasets (Supplementary Fig. 2). We used the results from the fixed-effect meta-analysis to perform further analyses (Supplementary Data 1).

### Mendelian randomisation
Instruments were selected from within or near the *SOST* gene, the gene encoding the drug target. A total of 5,769 meta-analysed single nucleotide polymorphisms (SNPs) were within ±500 kb from the *SOST* gene (chr17: 43,253,738–44,258,791 (build GRCh38/hg38)). As we focused on the selection of variants to a single region of the genome, a *p*-value threshold of $1 \cdot 10^{-6}$ was prespecified[26], which is distinct from the typical GWAS significance threshold: $5 \cdot 10^{-8}$. There were 54 SNPs that reached significance. We used pruning to select low-correlated variants ($r^2 \leq 0.3$), and obtained 2 SNPs (rs7220711, rs66838809) strongly associated with sclerostin levels (Table 1, Supplementary Fig. 1). Results of the meta-analysis for these two SNPs can be found in Supplementary Table 1. Both variants had an F statistic > 10, suggesting no weak instrument bias. The linkage disequilibrium (LD) matrix used to account for correlation between instruments can be found in Supplementary Table 2.

We extracted the genetic associations for the 2 instruments from each GWAS summary statistics of the positive control outcomes (Table 2). Both alleles were associated with increasing levels of heel BMD (rs7220711: Beta = 0.039, 95%CI = [0.035, 0.042] respect allele G; rs66838809: Beta = 0.073 95%CI = [0.067, 0.079] respect allele A), and with decreasing fracture risk (rs7220711: OR = 0.932 [0.905, 0.960]; rs66838809: OR = 0.866 [0.815, 0.920]) (See Supplementary Fig. 3). Mendelian randomisation results confirmed that sclerostin-lowering alleles were associated with higher heel BMD, with a beta of 1.00 (0.92, 1.08), and with lower hip fracture risk, with an OR of 0.16 [0.08, 0.30] per each SD decrease in sclerostin levels (Supplementary Table 3).

We extracted the genetic associations for the 2 SNPs to be used as instruments for all the GWAS summary statistics of the outcomes of interest (Table 2). Genetically predicted lower levels of sclerostin were associated with lower levels of HDL cholesterol, with beta = −0.15 [−0.2, −0.09], higher odds of coronary artery disease, with OR = 1.25 [1.01, 1.55] and higher odds of type 2 diabetes, with OR = 1.45 [1.11, 1.90]. Results for myocardial infarction and hypertension were borderline significant (OR = 1.35 [0.98, 1.87], and OR = 1.03 [0.99, 1.07], respectively). See Supplementary Table 4 for more details.

From the 502,396 UK Biobank participants recruited at baseline, 276,172 met our selection criteria and were included in the analysis (Supplementary Fig. 4). Baseline characteristics of these participants can be found in Supplementary Table 5.

We adjusted a multivariant linear regression between the instruments and some biomarkers from the UK Biobank biochemistry panel data (Table 2). MR results suggested that sclerostin-lowering alleles were associated with decrease levels of HDL cholesterol (Beta = −0.13 [−0.24, −0.02]) and increased levels of triglycerides (Beta = 0.26, [0.15, 0.38]) (Supplementary Table 6). Results for Apolipoprotein-B were borderline significant (Beta = 0.12 [−0.00, 0.24]).

A multivariant logistic regression was regressed on the categorical outcomes from UK Biobank (Table 2). *Cis*-MR results showed an increased risk of coronary artery disease (OR = 1.85 [1.12, 3.06]) and type 2 diabetes (OR = 1.62 [1.05, 2.51]) (Supplementary Table 6).

A multivariate cox regression was used to regress the survival outcomes (Table 2). Results suggested an increased risk of coronary artery disease (OR = 1.77 [1.10, 2.85]), and a borderline significant association with type 2 diabetes (OR = 1.67 [0.95, 2.92]) (Supplementary Table 6).

A forest plot with all the MR results is shown in Fig. 1.

### Colocalization
The colocalization analysis showed a strong overlap between the analysed *SOST* region and the positive control outcomes, with a colocalization probability >99% for both BMD and hip fracture risk (See Table 3). Colocalization with HDL cholesterol showed strong evidence supporting H3 hypothesis, which states that there is an association between both traits, but both are associated with distinct

**Table 1 | Single nucleotide polymorphisms (SNPs) employed as instruments for sclerostin levels in the Mendelian randomisation analysis**

| SNP | Effect allele | Other allele | Effect allele frequency | Beta (95% CI) | SE | P-Value | N | Position (GRCh38/Hg38) | $I^2$ | Q | Q P-value | F statistic |
|---|---|---|---|---|---|---|---|---|---|---|---|---|
| rs7220711 | G | A | 0.33 | −0.04 (−0.06, −0.03) | 0.01 | $1.11 \cdot 10^{-9}$ | 49,372 | 17:43712597 | 0.64 | 5.56 | 0.06 | 37.18 |
| rs66838809 | A | G | 0.09 | −0.07 (−0.09, −0.04) | 0.01 | $7.76 \cdot 10^{-9}$ | 49,372 | 17:43721253 | 0.74 | 7.61 | 0.02 | 33.38 |

Effect sizes were calculated using fixed-effects method. SNP Single nucleotide polymorphism, SE Standard error, Heterogeneity test ($I^2$ statistics), Q (Cochran's Q statistics), Q p-value).

genetic variants. Colocalization probabilities for other traits suggested that in this genetic region, only an association with sclerostin can be found (Supplementary Fig. 5).

### Stepwise pruning
We repeated the selection of instruments using different $r^2$ thresholds (0.001, 0.1, 0.5, 0.8). Clumping using an $r^2 \leq 0.001$ and $r^2 \leq 0.1$ gave the same result (1 variant: rs7220711). Instruments selected when using an $r^2 \leq 0.5$ gave the same selection of variants as in the main analysis. When clumping using $r^2 \leq 0.8$, 6 variants were selected: rs9910625, rs7220711, rs66838809, rs7213935, rs80107551, rs6503468.

Results using an $r^2 \leq 0.001$ were consistent with the main results, although we did not find a significant association with type 2 diabetes (either for the categorical and the survival outcome) in the UK Biobank cohort (Supplementary Fig. 6, Supplementary Table 7).

Positive control outcomes' results for $r^2 \leq 0.8$ were aligned with the main results. For GWAS outcomes, no association was seen with HDL cholesterol levels, coronary artery disease or type 2 diabetes. For UK Biobank outcomes, results only supported an association with increased levels of triglycerides and an increased risk of type 2 diabetes (either for the categorical and the survival outcomes) (Supplementary Fig. 7, Supplementary Table 9).

### Random-effects method
We used random-effects results to select the variants to be used as instruments. Following the same selection criteria as in the main analysis, we obtain a single SNP to be used as instrument: rs2342312.

Results using rs2342312 as instruments showed similar results as in the main analysis for positive controls and GWAS outcomes (Supplementary Table 9). Most UK Biobank outcomes were also aligned with the same results. However, random-effects results showed an association between lower levels of sclerostin and decreased levels of apolipoprotein-A (beta = −0.22 [−0.38, −0.07]), and with increased risk of myocardial infarction (OR = 2.67 [1.16, 6.1]; HR = 2.62 [1.17, 5.84]). An association with type 2 diabetes was not seen (See Supplementary Fig. 8).

### PCA analysis
We used principal component analysis to compute independent linear combinations of *cis*-variants associated with sclerostin. Afterwards, we used the first *k* principal components (or linear combinations) that explained more than 99.9% of the genetic variance observed. MR results were all aligned with the main findings. However, the association with coronary artery disease was borderline significant (Supplementary Table 9, Supplementary Fig. 9).

### Survival outcomes since UK Biobank enrolment
A survival analysis starting follow-up from first assessment in UK Biobank and excluding people with a previous cardiovascular event was conducted to approximate the target population potentially treatable with sclerostin inhibitor therapy in Europe. All the results showed the same direction as the main ones, but confidence intervals were much larger compared to the ones in the main analysis (Supplementary Table 9, Supplementary Fig. 9).

## Discussion
By meta-analysing three different cohorts, we created the largest GWAS summary statistics of genetically predicted sclerostin. We triangulated evidence using genetics to confirm large trial findings of an improved bone mineral density and great reduction in hip fracture risk in subjects with lowered levels of sclerostin. These findings were obtained from the most up to date GWAS studies for the outcomes.

We report on genetic evidence supporting that lower sclerostin levels are causally related with both coronary artery disease and type 2 diabetes. This was confirmed using either GWAS and UK Biobank data.

**Table 2 | Single nucleotide polymorphisms effect on the outcomes**

| Outcome | Source of data for the outcome | SNP | N | Effect allele | Effect allele frequency | Type of estimate | Estimate | Standard error | P-value |
|---|---|---|---|---|---|---|---|---|---|
| Heel bone mineral density | GWAS | rs7220711 | 426824 | G | 0.39 | Beta | 0.039 | 0.002 | 3.90E-70 |
| Heel bone mineral density | GWAS | rs66838809 | 426824 | A | 0.08 | Beta | 0.073 | 0.003 | 1.00E-85 |
| Hip fracture | GWAS | rs7220711 | 735354 | G | 0.42 | OR | 0.932 | 0.015 | 2.35E-06 |
| Hip fracture | GWAS | rs66838809 | 735354 | A | 0.09 | OR | 0.866 | 0.031 | 2.59E-06 |
| LDL cholesterol | GWAS | rs7220711 | 1226841 | G | 0.39 | Beta | 0.001 | 0.001 | 4.52E-01 |
| LDL cholesterol | GWAS | rs66838809 | 1228508 | A | 0.08 | Beta | 0.001 | 0.003 | 8.39E-01 |
| HDL cholesterol | GWAS | rs7220711 | 1244439 | G | 0.39 | Beta | −0.007 | 0.001 | 9.63E-07 |
| HDL cholesterol | GWAS | rs66838809 | 1241791 | A | 0.08 | Beta | −0.007 | 0.003 | 5.00E-03 |
| Fasting glucose | GWAS | rs7220711 | 177305 | G | 0.42 | Beta | 0 | 0.002 | 7.42E-01 |
| Fasting glucose | GWAS | rs66838809 | 178455 | A | 0.08 | Beta | 0.006 | 0.004 | 7.27E-02 |
| HbA1c | GWAS | rs7220711 | 132400 | G | 0.42 | Beta | 0.001 | 0.001 | 4.70E-01 |
| HbA1c | GWAS | rs66838809 | 132400 | A | 0.08 | Beta | 0.001 | 0.003 | 9.81E-01 |
| Coronary artery disease | GWAS | rs7220711 | 1143140 | G | 0.39 | OR | 1.011 | 0.005 | 3.38E-02 |
| Coronary artery disease | GWAS | rs66838809 | 1152860 | A | 0.08 | OR | 1.011 | 0.009 | 2.62E-01 |
| Myocardial infarction | GWAS | rs7220711 | 638717 | G | 0.4 | OR | 1.01 | 0.008 | 1.74E-01 |
| Myocardial infarction | GWAS | rs66838809 | 638717 | A | 0.08 | OR | 1.028 | 0.015 | 6.70E-02 |
| Ischaemic stroke | GWAS | rs7220711 | 1847683 | G | 0.41 | OR | 1.012 | 0.007 | 1.02E-01 |
| Ischaemic stroke | GWAS | rs66838809 | 1847683 | A | 0.08 | OR | 0.996 | 0.014 | 7.78E-01 |
| Hypertension | GWAS | rs7220711 | 462933 | G | 0.39 | OR | 1.001 | 0.001 | 5.40E-01 |
| Hypertension | GWAS | rs66838809 | 462933 | A | 0.08 | OR | 1.003 | 0.002 | 7.50E-02 |
| Type 2 diabetes mellitus | GWAS | rs7220711 | 933970 | G | 0.39 | OR | 1.019 | 0.007 | 4.58E-03 |
| Type 2 diabetes mellitus | GWAS | rs66838809 | 933970 | A | 0.08 | OR | 1.017 | 0.012 | 1.46E-01 |
| Cholesterol | UK Biobank | rs7220711 | 256586 | G | 0.38 | Beta | 0.002 | 0.003 | 5.45E-01 |
| Cholesterol | UK Biobank | rs66838809 | 256167 | A | 0.08 | Beta | 0.004 | 0.005 | 5.00E-01 |
| LDL cholesterol | UK Biobank | rs7220711 | 256105 | G | 0.38 | Beta | 0.002 | 0.003 | 4.14E-01 |
| LDL cholesterol | UK Biobank | rs66838809 | 255690 | A | 0.08 | Beta | 0.005 | 0.005 | 3.77E-01 |
| HDL cholesterol | UK Biobank | rs7220711 | 234948 | G | 0.38 | Beta | −0.006 | 0.003 | 2.66E-02 |
| HDL cholesterol | UK Biobank | rs66838809 | 234554 | A | 0.08 | Beta | −0.007 | 0.005 | 1.69E-01 |
| Triglycerides | UK Biobank | rs7220711 | 256375 | G | 0.38 | Beta | 0.012 | 0.003 | 3.00E-05 |
| Triglycerides | UK Biobank | rs66838809 | 255955 | A | 0.08 | Beta | 0.016 | 0.005 | 2.31E-03 |
| Apolipoprotein-A | UK Biobank | rs7220711 | 233627 | G | 0.38 | Beta | −0.004 | 0.003 | 1.98E-01 |
| Apolipoprotein-A | UK Biobank | rs66838809 | 233244 | A | 0.08 | Beta | −0.004 | 0.005 | 4.40E-01 |
| Apolipoprotein-B | UK Biobank | rs7220711 | 255322 | G | 0.38 | Beta | 0.004 | 0.003 | 1.36E-01 |
| Apolipoprotein-B | UK Biobank | rs66838809 | 254901 | A | 0.08 | Beta | 0.01 | 0.005 | 6.20E-02 |
| C-Reactive protein | UK Biobank | rs7220711 | 256040 | G | 0.38 | Beta | 0 | 0.003 | 1.00E + 00 |
| C-Reactive protein | UK Biobank | rs66838809 | 255625 | A | 0.08 | Beta | 0.004 | 0.005 | 4.77E-01 |
| Lipoprotein (a) | UK Biobank | rs7220711 | 203995 | G | 0.38 | Beta | −0.001 | 0.003 | 8.41E-01 |
| Lipoprotein (a) | UK Biobank | rs66838809 | 203682 | A | 0.08 | Beta | −0.004 | 0.006 | 4.55E-01 |
| Glucose | UK Biobank | rs7220711 | 234795 | G | 0.38 | Beta | 0.003 | 0.003 | 2.94E-01 |
| Glucose | UK Biobank | rs66838809 | 234409 | A | 0.08 | Beta | 0.008 | 0.005 | 1.43E-01 |
| HbA1c | UK Biobank | rs7220711 | 256419 | G | 0.38 | Beta | −0.001 | 0.003 | 7.50E-01 |
| HbA1c | UK Biobank | rs66838809 | 255984 | A | 0.08 | Beta | 0.003 | 0.005 | 5.42E-01 |
| Coronary artery disease | UK Biobank (categorical outcome) | rs7220711 | 269168 | G | 0.38 | OR | 1.027 | 0.012 | 2.79E-02 |
| Coronary artery disease | UK Biobank (categorical outcome) | rs66838809 | 268712 | A | 0.08 | OR | 1.039 | 0.022 | 8.68E-02 |
| Myocardial infarction | UK Biobank (categorical outcome) | rs7220711 | 269168 | G | 0.38 | OR | 1.028 | 0.015 | 5.71E-02 |
| Myocardial infarction | UK Biobank (categorical outcome) | rs66838809 | 268712 | A | 0.08 | OR | 0.983 | 0.027 | 5.34E-01 |

**Table 2 (continued) | Single nucleotide polymorphisms effect on the outcomes**

| Outcome | Source of data for the outcome | SNP | N | Effect allele | Effect allele frequency | Type of estimate | Estimate | Standard error | P-value |
|---|---|---|---|---|---|---|---|---|---|
| Ischaemic stroke | UK Biobank (categorical outcome) | rs7220711 | 269168 | G | 0.38 | OR | 1.003 | 0.024 | 9.11E-01 |
| Ischaemic stroke | UK Biobank (categorical outcome) | rs66838809 | 268712 | A | 0.08 | OR | 1.036 | 0.044 | 4.18E-01 |
| Hypertension | UK Biobank (categorical outcome) | rs7220711 | 269168 | G | 0.38 | OR | 1.004 | 0.006 | 4.84E-01 |
| Hypertension | UK Biobank (categorical outcome) | rs66838809 | 268712 | A | 0.08 | OR | 1.018 | 0.011 | 1.18E-01 |
| Type 2 diabetes | UK Biobank (categorical outcome) | rs7220711 | 269168 | G | 0.38 | OR | 1.018 | 0.011 | 8.82E-02 |
| Type 2 diabetes | UK Biobank (categorical outcome) | rs66838809 | 268712 | A | 0.08 | OR | 1.038 | 0.019 | 5.14E-02 |
| Coronary artery disease | UK Biobank (survival outcome) | rs7220711 | 269168 | G | 0.38 | HR | 1.025 | 0.012 | 3.16E-02 |
| Coronary artery disease | UK Biobank (survival outcome) | rs66838809 | 268712 | A | 0.08 | HR | 1.036 | 0.021 | 9.14E-02 |
| Myocardial infarction | UK Biobank (survival outcome) | rs7220711 | 269065 | G | 0.38 | HR | 1.027 | 0.014 | 5.60E-02 |
| Myocardial infarction | UK Biobank (survival outcome) | rs66838809 | 268608 | A | 0.08 | HR | 0.984 | 0.026 | 5.49E-01 |
| Ischaemic stroke | UK Biobank (survival outcome) | rs7220711 | 269168 | G | 0.38 | HR | 1.002 | 0.024 | 9.21E-01 |
| Ischaemic stroke | UK Biobank (survival outcome) | rs66838809 | 268712 | A | 0.08 | HR | 1.037 | 0.043 | 4.02E-01 |
| Hypertension | UK Biobank (survival outcome) | rs7220711 | 264485 | G | 0.38 | HR | 1.003 | 0.005 | 5.48E-01 |
| Hypertension | UK Biobank (survival outcome) | rs66838809 | 264045 | A | 0.08 | HR | 1.012 | 0.009 | 1.71E-01 |
| Type 2 diabetes | UK Biobank (survival outcome) | rs7220711 | 259515 | G | 0.38 | HR | 1.019 | 0.014 | 1.67E-01 |
| Type 2 diabetes | UK Biobank (survival outcome) | rs66838809 | 259093 | A | 0.08 | HR | 1.041 | 0.025 | 1.02E-01 |

Notice that all the estimates are reported respect the effect allele. Statistical methods used to calculate GWAS outcomes are reported in the original publications. UK Biobank outcomes' effect sizes were calculated using linear, logistic and cox regression for continuous, categorical and survival outcomes, respectively. *SNP* Single nucleotide polymorphism.

We also highlight a potential (but borderline significant in our study) causal relationship between genetically predicted lower levels of sclerostin and increased risk of myocardial infarction and hypertension. Our study does not show any association between sclerostin levels and ischaemic stroke.

More importantly, our analysis of UK Biobank shows a causal relationship between sclerostin inhibition on key cardiovascular biomarkers, including an inverse association between sclerostin levels and the levels of triglycerides, and (potentially) apolipoprotein B. Sclerostin levels were directly associated with protective biomarkers like HDL cholesterol in the MR analysis. However, colocalization results suggest that the association seen with HDL cholesterol could be due to confounding by linkage disequilibrium.

Sensitivity analysis showed that the direction of all our signals was robust. More importantly, when selecting instruments from random-effects meta-analysed dataset, results suggested a causal relationship between lower levels of sclerostin and lower levels of apolipoprotein-A, and an increased risk of myocardial infarction.

Additionally, and in line with these findings, lower levels of sclerostin appeared to be causally associated with higher levels of triglycerides in UK Biobank, and a higher risk of type 2 diabetes mellitus in both the meta-analysis of GWAS and our analysis of UK Biobank. Type 2 diabetes is a well-established risk factor for ischaemic heart disease.

Large pivotal randomised controlled trials of two sclerostin inhibitors have demonstrated a positive effect of these treatments, leading to improvements of increased bone mineral density[14,27]. Although blosozumab was discontinued, RCT data on anti-fracture effects are available on the only approved sclerostin inhibitor for human use: romosozumab, which was shown to be superior to placebo[17], and better than first line therapy alendronate in reducing fracture risk[16]. Our findings support and triangulate this evidence by providing compelling genetic evidence of a positive effect of lower sclerostin levels on bone health.

More intriguing is the available RCT data on the cardiovascular safety of sclerostin inhibitors. While no effect was observed in the large phase 3 placebo-controlled trial[14], an increase in CVD outcomes was observed in a smaller trial in men[17] and in a head-to-head RCT versus alendronate[16]. Previous cohort studies have not confirmed these findings, with conflicting results, possibly due to confounding[28]. Our analyses confirm a causal relationship between lifelong effects of lower levels of sclerostin and an excess risk of coronary artery disease, both in meta-analysis of previous GWAS as well as in UK Biobank participants. These results are in line with previous genetic evidence[19,29].

At the time of approval of romosozumab, little was known on possible mechanisms that could cause an increased cardiovascular risk with this therapy. Smaller studies have suggested a potential association with artery calcification[30] and with angiogram-measured coronary disease severity[28] but little was known on other cardiovascular biomarkers. A previous study[20], which also performed a *cis*-MR approach using a meta-analysis of sclerostin levels in Fenland and Interval studies, reported comparable findings regarding the biomarkers. Despite the wide confidence interval (including the null), their direction of effect suggests increased triglycerides and apolipoprotein B, and reduced levels of protective HDL and apolipoprotein A. Similarly, they found a higher risk of type 2 diabetes mellitus among UK Biobank

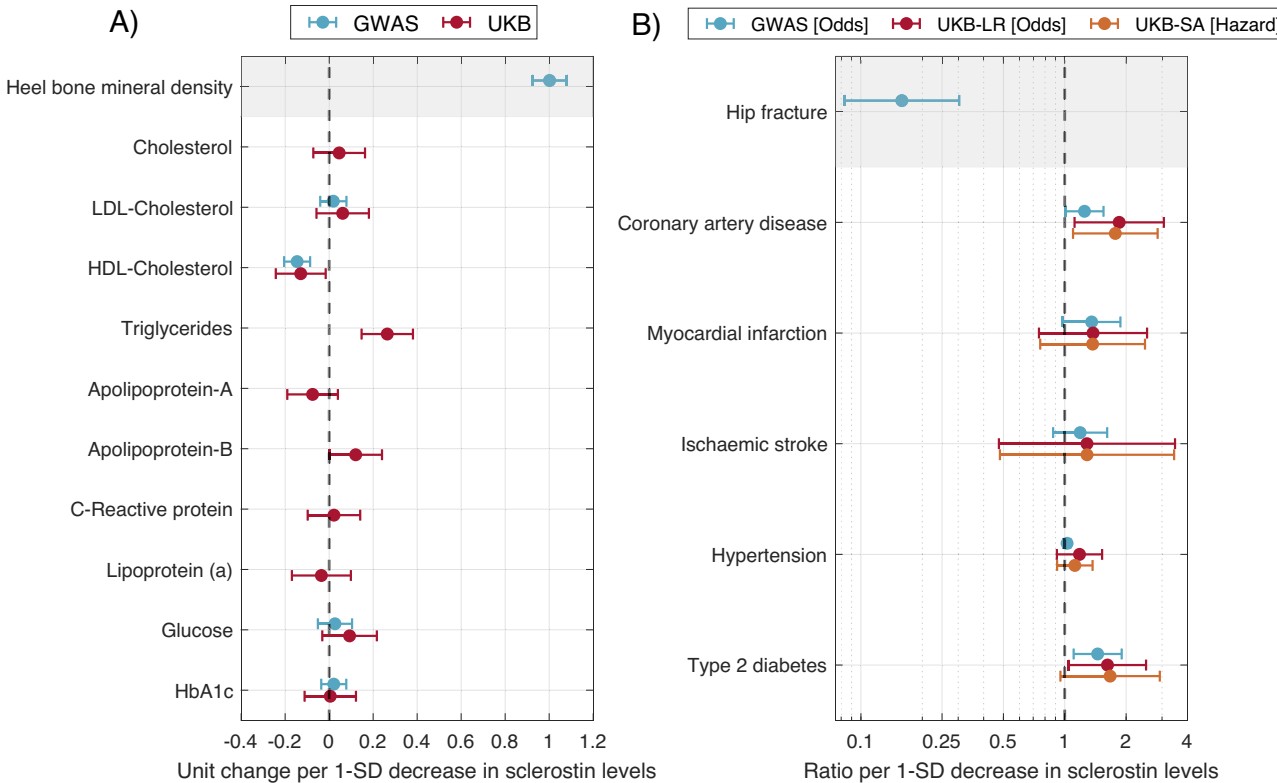

**Fig. 1 | Forest plot of MR estimates.** Effect sizes were calculated using the generalised inverse variance weighted method. Blue represents the results based on published GWAS summary statistics as the outcome; maroon the results obtained using a linear regression (for continuous outcomes) and logistic regression (for categorical outcomes) on the UK Biobank outcomes; and orange indicates the results using cox regression for UK Biobank survival outcomes. Error bars indicate the 95% confidence interval. **A** The horizontal axis shows the SD change per 1 SD decrease in sclerostin levels. GWAS results for LDL and HDL-Cholesterol are in mg/ dL, mmol/L for GWAS results of fasting glucose, % change for HbA1c and SD increase for the other outcomes. **B** The horizontal axis shows the odds/hazard ratio per 1 SD decrease in sclerostin levels. Source data are provided as a Source Data file. Note: UKB-LR = UK Biobank logistic regression, UKB-SA = UK Biobank survival analysis. Details about sample sizes used to calculate the MR estimates can be found in Supplementary Table 5, Supplementary Table 7, and Supplementary Table 8.

**Table 3 | Colocalization results**

| Outcome | N SNPs | Prior probabilities (%) | | Posterior probabilities (%) | | | | | | |
|---|---|---|---|---|---|---|---|---|---|---|
| | | P1 | P2 | P12 | P(H0) | P(H1) | P(H2) | P(H3) | P(H4) | |
| Heel bone mineral density | 171 | 0.01 | 0.01 | 0.001 | 0.00 | 0.00 | 0.02 | 0.26 | 99.72 | |
| Hip fracture | 108 | 0.01 | 0.01 | 0.001 | 0.00 | 0.02 | 0.02 | 0.23 | 99.72 | |
| LDL cholesterol | 208 | 0.01 | 0.01 | 0.001 | 5.10 | 94.24 | 0.02 | 0.43 | 0.21 | |
| HDL cholesterol | 208 | 0.01 | 0.01 | 0.001 | 0.00 | 0.00 | 5.13 | 94.79 | 0.07 | |
| Glucose | 149 | 0.01 | 0.01 | 0.001 | 4.89 | 90.02 | 0.04 | 0.71 | 4.34 | |
| HbA1c | 149 | 0.01 | 0.01 | 0.001 | 5.09 | 93.77 | 0.03 | 0.47 | 0.64 | |
| Coronary artery disease | 197 | 0.01 | 0.01 | 0.001 | 5.03 | 92.69 | 0.06 | 1.01 | 1.21 | |
| Myocardial infarction | 99 | 0.01 | 0.01 | 0.001 | 4.88 | 89.89 | 0.01 | 0.10 | 5.12 | |
| Ischaemic stroke | 102 | 0.01 | 0.01 | 0.001 | 5.10 | 93.90 | 0.01 | 0.13 | 0.86 | |
| Hypertension | 121 | 0.01 | 0.01 | 0.001 | 5.14 | 94.68 | 0.00 | 0.03 | 0.14 | |
| Type 2 diabetes | 120 | 0.01 | 0.01 | 0.001 | 4.91 | 90.30 | 0.19 | 3.58 | 1.02 | |

P1 corresponds to the prior probability of each variant being associated with sclerostin. P2 corresponds to the prior probability of each variant being associated with the outcome. P12 corresponds to the probability of each variant being associated with both, sclerostin and the outcome. P(H0) corresponds to the probability that the null hypothesis is true. P(H1) is the probability that there is an association with sclerostin but not with the outcome. P(H2) is the probability that there is an association with the outcome but not with sclerostin. P(H3) is the probability that in this region there is an association with both but because of different genetic variants. P(H4) is the probability that there is an association with both traits at a shared causal variant.

participants with lower predicted levels of sclerostin. However, Zheng et al. did not find evidence of an association with coronary artery disease, whereas we observed an increase on its risk. Their results also suggested an increased risk with myocardial infarction, whereas in our study this association is borderline significant. A previous meta-analysis of genetic studies found similar associations[19].

In addition to the *cis*-effect of the genetic variants in the *SOST* gene, variants in other genes (i.e., *trans*) have also been associated with circulating levels of sclerostin. Recent findings using mice models found a strong association between *B4GALNT3* gene (loci in chromosome 12) and sclerostin levels[31]. The study revealed that *B4GALNT3* was co-expressed with *SOST* gene. By using variants in the *B4GALNT3* to

genetically predict reduced levels of sclerostin, they found a decrease in BMD and an increase in risk of fracture, but no association was observed with myocardial infarction or stroke. However, our study, minimise the potential bias due to horizontal pleiotropy by using genetic variants in *cis*.

Our study has several strengths that enhance the reliability of our findings. First, we included the use of large-scale genetic data, enabling us to conduct a Mendelian randomisation study that triangulates evidence. Second, we minimized the risk of horizontal pleiotropy by restricting the selection of variants within or close the *SOST* gene. Third, we employed positive control outcomes to validate our instruments; and fourth, we used different data sources (i.e., summary statistics and patient-level data) to enhance the robustness of our analysis.

Although our study has many strengths there are some limitations that need to be taken into consideration. Firstly, in the meta-analysis of sclerostin levels we found some statistical evidence of heterogeneity across the three sclerostin cohorts in some of the SNPs used as genetic instruments for this study. This may introduce bias in our results, underestimating the standard errors of the effect sizes. However, we decided to use the fixed-effect results to increase the statistical power and include random-effects methods as a sensitivity analysis. Second, MR studies the effect of small lifelong effects of changes in circulating levels of sclerostin, whereas pharmacological treatments are administered for a shorter time and at a specific point in time. Thus, the effect estimates from this study should not be interpreted to be the effect of the pharmacological intervention, but to be a useful indication of presence and direction of causal effects[24,32]. Third, we have assumed linearity between sclerostin levels and all the outcomes, but some literature suggest that this may not be completely true[28]. Further research is needed to provide more insights to this assumption. Fourth, colocalization showed limited evidence of a common causal effect, probably due to low statistical power[33]. Fifth, only participants with European ancestry have been included in this analysis. Hence, results from this study cannot be generalised to other populations with different genetic background.

We have genetically validated the association between low levels of sclerostin and improved bone health and reduced fracture risk. Our analyses provide evidence of a causal relationship between reduced sclerostin and an increased risk of coronary artery disease and myocardial infarction. Our findings suggest that alterations in triglycerides and lipid metabolism could at least partially explain the potential negative effects of sclerostin inhibition on cardiovascular outcomes.

## Methods

### Study design and data sources
We have conducted a two-sample *cis*-Mendelian randomisation study. For the exposure, we meta-analysed three GWAS summary statistics to extract SNPs associated with sclerostin[34–36]. For the outcome, two types of data sources were used: publicly available GWAS summary statistics and single SNP person level data from UK Biobank (Fig. 2).

### Exposure
The three GWAS summary statistics for circulating protein levels of sclerostin were obtained from three cohorts: the INTERVAL Study ($N = 3301$)[34], from deCODE genetics ($N = 35,559$)[35], and the Fenland Study ($N = 10,708$)[36] and downloaded from https://app.box.com/s/u3flbp13zjydegrxjb2uepagp1vb6bj2, https://www.decode.com/summarydata, and https://omicsscience.org/apps/pgwas, respectively. Supplementary Table 7 summarises the characteristics of each cohort. All protein levels were measured using the somalogic SomaScan platform. All sclerostin measures were reported in standard deviation (SD) units.

In the INTERVAL study, age, sex, duration between blood draw and processing (binary, ≤ 1 day / >1 day) and the three first principal components (PCs) were used as covariates for the model. The Fenland study used age, sex and the first 10 PCs to adjust the model. The study from deCode genetics only adjusted for age, sample age and sex.

### Outcomes
**Positive control outcomes.** We used 2 publicly available GWAS summary statistics for our positive control outcomes: estimated heel bone mineral density (BMD)[37] and hip fracture[38] (Supplementary Table 8). We define positive control outcomes as outcomes that are causally influenced by our exposure, for which the causal effect's direction is previously known. In this case, we expected to see a causal relationship between genetically predicted lower sclerostin and an increase in heel BMD levels and a decrease in hip fracture risk. Hence, these outcomes help us to ensure that the methodology followed is reliable.

Effect sizes for BMD GWAS were given in SD units, whereas for the GWAS of hip fracture were provided in log(OR).

**Outcomes from GWAS summary statistics.** We used 9 other GWAS summary statistics of the following outcomes (Supplementary Table 8): cardiovascular events (coronary artery disease[39], myocardial infarction[40], ischaemic stroke[41]); risk factors (hypertension[42], type 2 diabetes mellitus[43]); and biomarkers (LDL cholesterol[44], LDL cholesterol[44], fasting glucose[45], HbA1c[45]).

Effect sizes for quantitative data (LDL cholesterol, HDL cholesterol, fasting glucose and HbA1c) were given in mg/dL, mg/dL, mmol/L and % unit change, respectively. Categorical data (cardiovascular events, and risk factors) were provided in log(OR).

**Outcomes from UK Biobank.** UK Biobank is a large prospective study with over 500,000 participants aged 40-69 years old when recruited in 2006–2010[46]. For phenotypes used in this study we used the information collected via questionnaires, physical measurements, prescriptions, and interviews in baseline assessments. We also used linked data from General Practice (GP) (with follow-up data until August 2019) and Hospital Episodes Statistics (HES) (with follow-up data until November 2021).

Records from GP data with invalid date were excluded. Records from HES happening at 01/01/2020 or later were also removed.

For this study we restricted the cohort to participants from UK Biobank with European ancestry, with the same sex and genetic sex recorded, with no sex chromosome aneuploidy and without kinship with other participants. Other exclusion criteria for specific analysis are stated below. Details on genotyping, imputation and quality control data of UK Biobank patients have been described previously[47].

We studied the following 15 outcomes present in UK Biobank: cholesterol, LDL and HDL cholesterol, apolipoproteins A and B, lipoprotein (a), C-reactive protein, triglycerides, glucose, and HbA1c as biomarkers; coronary artery disease, myocardial infarction, and ischaemic stroke as cardiovascular events; and hypertension and type 2 diabetes as risk factors. The phenotyping codes used for the UK Biobank outcomes and where they were identified (initial assessment, HES, or GP) can be found in Supplementary Table 10.

### Statistical methods
**Meta-analysis.** We conducted a meta-analysis of three GWAS of circulating sclerostin levels. Only SNPs within chromosome 17 were included in the meta-analysis, as this is the chromosome where the *SOST* gene is located. Multiallelic SNPs, variants with invalid IDs, with the same allele for "effect" and "other", with alleles "D" or "I", with alleles frequency ≤ 0 or ≥ 1, P-values < 0 or > 1, and with standard errors ≤ 0 or equal to infinity, were removed. Additionally, all alleles were standardised to a reference allele using software GWAMA[48] and METAL[49].

Fixed-effects inverse-variance weighted method was used to combine the three GWAS results in the meta-analysis. We also used $I^2$,

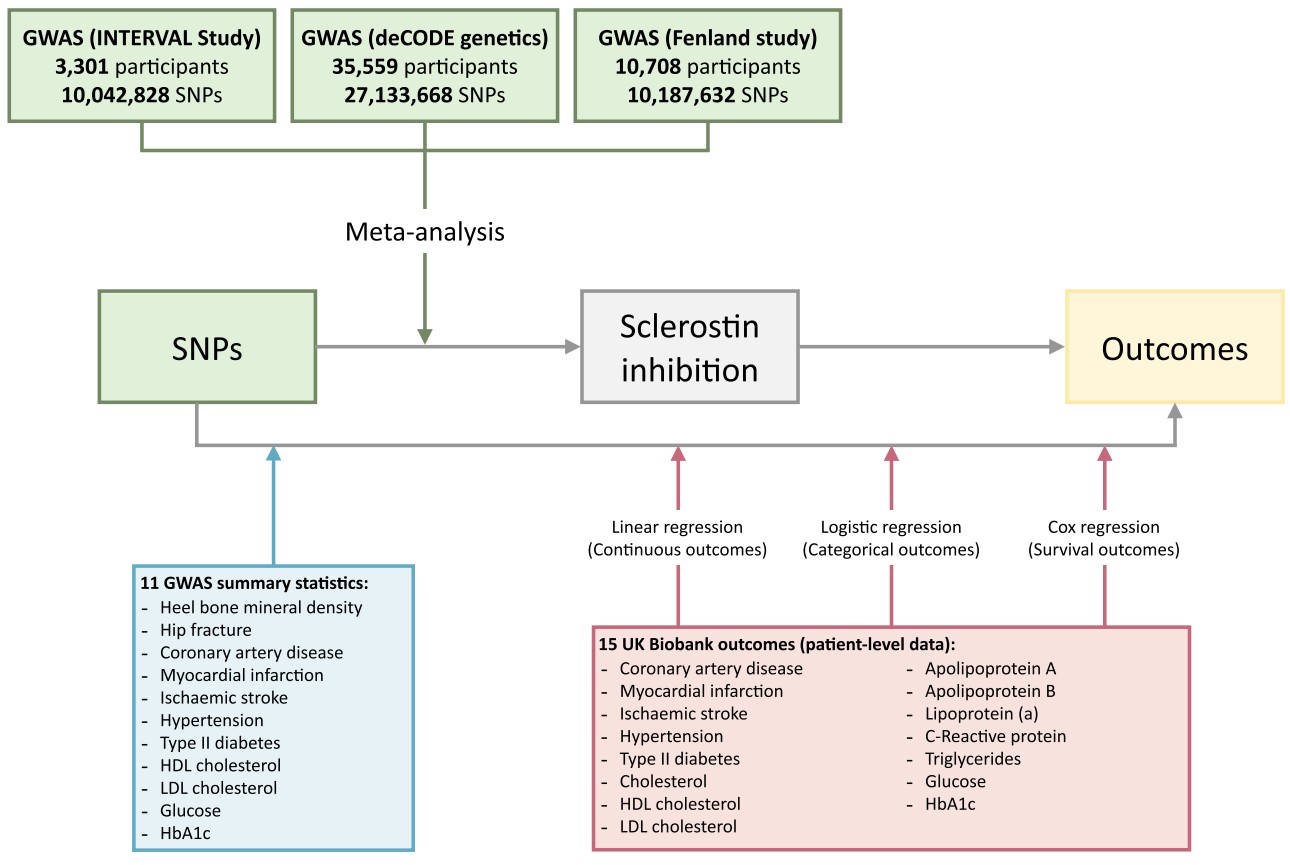

**Fig. 2 | Schema of the study design.**

Cochran's Q statistics and Cochran's Q P-value to assess heterogeneity. We calculated the genomic inflation factor to assess whether there was genetic bias within our fixed-effects meta-analysed dataset.

As sclerostin GWAS were configured in different assemblies, we used *biomaRt*[50] package to assign Hg38/GRCh38 positions to the meta-analysed variants. Later, we selected only those variants within 500 kb from the start and end of the *SOST* gene.

**Mendelian randomisation.** The combined effect sizes from the 3 GWAS meta-analysis were used as exposures for our *cis*-MR analysis. The 11 publicly available GWAS and the 15 traits from UK Biobank were used as outcomes.

**Instrument selection criteria for the exposure.** In Mendelian randomisation, genetic variants must fulfil three assumptions to be considered valid instruments[51]: (1) they are associated with the exposure, (2) they cannot be associated with the outcome through confounding pathways, and (3) they can only affect the outcome through the exposure. To ensure that our instruments fulfilled these criteria, we first restricted the selection of variants within 500 kb from the *SOST* gene (*cis*-variants) to reduce the risk of horizontal pleiotropy. We also selected those SNPs strongly associated with sclerostin protein levels (meta-analysed p-value lower than $1 \cdot 10^{-6}$). This value was used after literature review of common thresholds used for *cis*-Mendelian randomisation approaches[52,53]. We pruned the genetic associations to remove SNPs in strong linkage disequilibrium (LD) ($r^2 > 0.3$, clumping window 500 kb) and hence, minimize Type I errors without sacrificing statistical power[54]. This threshold was chosen as recommended from previous literature of Mendelian Randomization[52,55], which suggest that using $r^2 > 0.3$ can result in substantial numerical instabilities. We used the 1000 genomes LD European reference panel to compute the LD matrix. We used F-statistic to evaluate the strength of each

instrument[56]. This resulted in a sclerostin genetic instrument based on strongly associated SNPs to study the effect of genetically predicted sclerostin levels on the outcomes.

**Positive control outcomes.** Effect sizes, allele frequencies, standard errors, and p-values for each instrument were extracted from the GWAS summary statistics of each positive control outcome[37,38]. MR estimates are reported as SD change/OR per 1 SD decrease in sclerostin levels for heel bone mineral density and hip fracture risk, respectively.

**Outcomes from publicly available GWAS.** We extracted the effect sizes, allele frequencies, standard errors, and p-values of the selected instruments from the 9 publicly available GWAS summary statistics of each outcome[39–45].

MR estimates for HDL and LDL cholesterol, fasting glucose and HbA1c were reported as mg/dL, mg/dL, mmol/l and % change per 1 SD decrease in sclerostin levels, respectively. MR estimates for categorical outcomes are reported as OR per 1 SD decrease in sclerostin levels.

**Outcomes from UK Biobank.** Patient-level data for continuous outcomes (i.e., biomarkers) were standardised to SD units. We performed a linear regression adjusting by sex (field 31), age at first assessment (field 21003), genetic batch (field 22000), and the first 10 PCs (field 22009). The 10 genetic PCs were used as reported from UK Biobank.

For categorical variables (i.e., cardiovascular events, and risk factors) we estimated the effect size of the genetic instruments on the outcomes through two different adjustments: I) a logistic regression; II) a cox regression using time to event since birth date[57,58]. In this last adjustment, we excluded patients with missing outcome date. In both fittings, sex, age at first assessment, genetic batch and the first 10 PCs were used as covariates.

The beta coefficients obtained from the different regressions were used later for computing the MR estimates. MR estimates for continuous outcomes were reported as SD change per 1 SD decrease in sclerostin levels, whereas for categorical and survival outcomes were reported as OR/HR per 1 SD decrease in sclerostin levels, respectively.

**Mendelian randomisation analysis.** First, we harmonized the data respect to the alleles of the LD matrix. The LD matrix was computed using PLINK[59] and the European 1000 genomes reference dataset. We used a generalised inverse-variance weighted (IVW) regression to account for correlation among instruments[54]. A *p*-value lower than 0.05 was used to determine statistical significance.

**Colocalization.** MR studies assume that the instruments influence the outcome only through the exposure[23]. To assess this assumption, we conducted colocalization analysis[28] within ±20 kb from the *SOST* gene. This statistical method evaluates whether the estimated levels of circulating sclerostin and the results may be causally impacted by the same genetic variants, or they may be influenced by distinct variants that are correlated with each other, leading to the violation of the MR assumptions.

We compared the variants' effect on sclerostin with their effect in positive control outcomes, cardiovascular events, risk factors, and biomarkers (i.e., from the 11 publicly available GWAS). In our study, we employed the default prior probabilities for colocalization: $P_1 = 10^{-4}$, $P_2 = 10^{-4}$, and $P_{12} = 10^{-5}$ and deemed a posterior probability for H4 greater than 80% as substantial evidence of colocalization[33].

**Sensitivity analyses**

**Stepwise pruning.** We used different $r^2$ thresholds ($r^2 \leq \{0.001, 0.1, 0.5, 0.8\}$) to clump our instruments and assess the robustness of our results[54]. Clumping window was maintained to 500 kb. Outcomes tested for association were the same ones included in the main analysis (positive controls, GWAS summary statistics, and UK Biobank outcomes). When there was more than one instrument, the generalised inverse variance weighted method was used for the MR analysis. When only one variant was obtained as instrument, the Wald ratio was used.

**Random-effects method.** Results from the random-effects method meta-analysis were used to assess if our results were consistent when considering heterogeneity between sclerostin GWAS. The main analysis was repeated but using instruments selected from the random-effects meta-analysed results for sclerostin GWAS. As we only found one variant as instrument, the Wald ratio was used to compute the MR estimates.

**PCA analysis.** Principal component analysis was first proposed by Burgess et al.[60] and later generalised to *cis*-Mendelian randomisation by Gatzanis et al.[54]. Briefly, this method computes linear combinations of genetic variants that are orthogonal to each other and that explain as much of the genetic variance observed. Later, these independent linear combinations are used as genetic instruments for the MR. An IVW method is used to estimate the MR effects. We implemented this method by selecting, first, those fixed-effects meta-analysed variants with a *p*-value $\leq 1 \cdot 10^{-6}$ and then, selecting the first *k* principal components that explain 99.9% of the variation in the genetic data.

**Survival outcomes since UK Biobank enrolment.** A survival analysis using cox regression with UK Biobank data was performed using time to event since UK Biobank first assessment date. Patients with events previous than the first assessment or missing outcome date were excluded. Sex, age at first assessment, genetic batch, and the first 10 PCs were used as covariates in the cox regression model when calculating instruments-outcomes effect size. More details about exclusion criteria were outcome specific and can be found in Supplementary Table 10. This is in line with the current label of the only approved sclerostin inhibitor (romosozumab) in Europe, which is contraindicated in patients with a previous history of myocardial infarction or stroke[61].

**Software/Implementation.** We used METAL (Version released on 25/03/2011)[49] and GWAMA (Version 2.2.2)[48] to perform the two different models of the meta-analysis. We used R version 4.2.2 and the packages *dplyr* (Version 1.1.3)[62] to perform the analysis, *biomaRt* (3.18)[50] to assign SNPs' position, *TwoSampleMR* (0.5.8)[63], and *MendelianRandomisation* (Version 0.9.0)[64] for the MR analysis, and *survival* package (3.5.7) to perform the cox regression. We also used the *coloc* package (5.2.3)[65] to perform the colocalization. Figures were created using MATLAB version R2023b. This manuscript has been written following the reporting guideline STROBE-MR[66].

## Data availability
All GWAS used are publicly available (see Supplementary Table 3 and Supplementary Table 4 for the access links). UK Biobank individual-level source data used in the study can be accessed by applying for access at http://ukbiobank.ac.uk/register-apply/. Source data are provided in this paper. Source data are provided with this paper.

## Code availability
All the codes used for the analyses can be found in the following publicly available GitHub repository: https://github.com/oxford-pharmacoepi/MR_Sclerostin.

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

## Acknowledgements

DPA receives funding from the UK National Institute for Health and Care Research (NIHR) in the form of a senior research fellowship. DPA's group received partial support from the Oxford NIHR Biomedical Research Centre. DG is supported by the British Heart Foundation Centre of Research Excellence (RE/18/4/34215) at Imperial College. DN is funded by the European Health Data & Evidence Network (EHDEN) and OPTIMA which has received funding from the Innovative Medicines Initiative 2 (IMI2) Joint Undertaking under grant agreement No 806968 and No. 101034347 respectively. IMI2 receives support from the European Union's Horizon 2020 research and innovation programme and European Federation of Pharmaceutical Industries and Associations (EFPIA). The sponsors of the study did not have any involvement in the writing of the manuscript or the decision to submit it for publication. JQX is funded through Jardine-Oxford Graduate Scholarship and a titular Oxford Clarendon Fund Scholarship. This research has been conducted using the UK Biobank Resource under Application Number 98358. The authors express their sincere gratitude to all the UK Biobank study participants for generously providing an invaluable resource to advance scientific research. The authors extend their appreciation to the UKBB management team for their dedication. This work uses data provided by patients and collected by the NHS as part of their care and support.

## Author contributions

Conceptualisation (MAH, JQ, DN, DG, DPA, APU); data curation (MAH); statistical analysis (MAH); supervision (JQ, DPA, CP, MC, APU); interpretation of data (MAH, JQ, MC, DPA, APU); draughting of the manuscript (MAH, DPA); critical revision of the manuscript (JQ, DN, CP, DG, MGM, DPA, APU). All authors reviewed and approved the final version. The views expressed in this article are the personal views of the author(s) and may not be understood or quoted as being made on behalf of or reflecting the position of the regulatory agency/agencies or organisations with which the author(s) is/are employed/affiliated.

## Competing interests

DPA's department has received grant/s from Amgen, Chiesi-Taylor, Lilly, Janssen, Novartis, and UCB Biopharma. His research group has received consultancy fees from Astra Zeneca and UCB Biopharma. Amgen, Astellas, Janssen, Synapse Management Partners and UCB Biopharma have funded or supported training programmes organised by DPA's department. The views expressed in this study are the personal views of MGM and do not represent the views of her current employer, the European Medicines Agency. All other authors declare no conflicts of interest.

### Ethics approval & consent to participate

All participants provided informed consent to participate in each one of the GWAS we have used. UK Biobank received ethical approval from the North West Multi-centre Research Ethics Committee (REC reference: 16/NW/0274). All participants provided informed consent to participate.
