## [Peer Review File · Nature Communications]

Reviewers' Comments:

Reviewer #1:

Remarks to the Author:

This paper reports findings from an MR study suggesting that a reduction in sclerostin level leads to an increased risk of MI and a number of CVD-related risk factors. This finding is potentially important as it adds to other evidence pointing to an increased CVD risk following treatment with sclerostin inhibitors, of which romosozumab is currently being marketed to treat osteoporosis. A similar study has recently been published (J Zheng et al, Arthritis and Rheumatology 2023 doi: 10.1002/art.42538), based on a slightly smaller sample size for the sclerostin GWAS providing the genetic instrument for circulating sclerostin. Zheng et al found similar effects on MI and T2DM risk, but null effects for other outcomes reported here. Specific points are as follows:-

1. Line 69: Having obtained the largest dataset to date of genetically predicted sclerostin levels, the authors focused on cis SNPs as genetic instruments, which makes sense as a strategy for reducing horizontal pleiotropy but would benefit from some explanation/justification.
2. The introduction could mention previous related MR studies; as well as the Zheng paper (included as a preprint here but important to cite the final version as there were some significant differences), papers by Bovijn et al Sci Transl Med 2020; 12:eaay6570 and G Holdsworth et al J Bone Miner Res 2021; 36: 1326–39 could also be cited.
3. Line 92 - a p values threshold of 10^{-6} was applied for selecting cis SNPs, which requires further justification (the convention is to apply 5×10^{-8})
4. Line 92 - "r" is used throughout instead of "r²"
5. Line 92 - r² of <0.8 was used for pruning. Use of a relatively high threshold has been suggested in these types of analyses in order to boost the number of instruments. However, unless steps are taken to adjust the IVW analysis for correlations between instruments, this approach underestimates the causal standard error. I suspect that many of the positive findings they report are due to relatively narrow confidence limits as they have failed to take account of correlation between instruments.
6. The structure of the manuscript is somewhat confusing given the second section refers to additional MR studies using UK Biobank data, whereas the first section also uses UK biobank data. Eg the heel BMD data in Table 2 is exclusively UK Biobank data, with similar analyses based on a subset of the data repeated in Table 3.
7. Line 233 and following - see above - findings from Zheng et al need to refer to the published paper rather than the preprint. Overlap of samples used in this meta-analyses with Zheng et al could also be mentioned (Interval and Fenland were used in both).
8. Line 270: I didn't quite understand this point - if this is about simulating different effects of sclerostin inhibition, this refers to a scaling function which doesn't affect

statistical power.

Reviewer #2:

Remarks to the Author:

Alcalde-Herraiz et al conducted a comprehensive analysis to estimate the effect of sclerostin inhibition on cardiometabolic diseases and risk factors using both Mendelian randomization (MR) and observational analysis. The findings suggested that sclerostin inhibition increased bone health but may increase T2D/glucose, lipids levels, and increase risk of CAD and MI. This aligns with some recent findings using similar pQTL MR setting (but slightly different exposure and some more outcomes). There are some sections that can be improved to reach the standard of Nature Communications.

Major comments:

1. Please provide a completed STROBE-MR checklist. The authors say they submitted it at the end of the methods section. It should be sent to the reviewers.
2. Meta-analysis of sclerostin GWAS in the cis region. Is there any standard QC been conducted before the meta-analysis? E.g. using some tools to standardize the alleles to a reference panel. Fixed effect meta-analysis results were reported and used for the MR. However, some SNPs showed evidence of heterogeneity (e.g. $I^2=0.74$ for rs66838809). What is the SNP effects of these SNPs with heterogeneity in each cohort? There is evidence to show that SOMA logic v3 and v4 have very different effect estimates. Have this been considered here? Or any other reasons why there are heterogeneity across studies?
3. The MR analysis using UK Biobank data. It is interesting to try to use survival analysis results as outcome. I think there are some GWAS methods, e.g. GATE that can be used, which avoiding some potential biases.
4. For the UK Biobank outcomes, it is unclear why we have to use UKBB outcomes? E.g. for lipids traits, we can use GLGC data, for glucose traits, we can use MAGIC data, which are measured after fasting. All these data have larger sample size, which may increase reliability and power.
5. The generalised IVW method. The confidence interval of the MR estimates seems to be narrow than other studies. I wonder if even small differences in LD between the discovery sample and the outcome sample in a two sample MR could lead to some double counting, if the signals are all coming from LD with a single non-measured causal variant.
6. Except the generalised IVW, some other methods that better control for weak instrument bias can be considered, e.g. cML-MA, debiased IVW and MRAID
7. The survival analysis, it is an interesting idea to do a survival analysis, which will be helpful for evidence triangulation. However, this section need more explanations. For

example, what is the exposure of this survival analysis? usage of sclerostin inhibitors? circulating sclerostin levels or PGS of sclerostin inhibition? what is the control group of this analysis? non-users? what is the average follow-up period? For covariates, why only sex and first 10 PCs were used? Please consider other potential confounding factors that may bias the results, e.g. study centre, age, usage of other drugs etc.

8. Colocalization, weak evidence was observed, which could be a power issue for the outcome data. or maybe too many cis signals in the same region? Will a suise+coloc pipeline be helpful here?

Minor comments:

1. Title is unclear, “cause-effect relationship” is vague. “UK Biobank analysis” is an observational analysis?
2. Abstract, causal association. Not a nice combination given the conceptual difference between causality and association.
3. Introduction, need to cite MR studies of sclerostin and explain the novelty of the current study clearly.

Reviewer #3:

Remarks to the Author:

This is a nice application of MR that investigates the relationship between sclerostin and a range of disease outcomes. This is a nice addition to the literature that goes considerably beyond other recent papers. While superficially similar, this paper is a considerable advance on Zheng et al, which is reflected in the definitive and precise results provided. For example, the current study estimates an odds-ratio for MI of 1.28 (95%CI: 1.18, 1.38) versus 1.35 (95%CI: 1.01-1.79) in Zheng et al. I.e. the current study has standard errors that are 3.6 fold smaller on the log-odds scale. This is roughly the equivalent to a 13 fold increase in sample size. It would be helpful for your readers to be explicit about where the additional power from your study comes from (the sample size for your discovery GWAS is slightly larger, so this is unlikely to explain it).

The study also presents additional novel data from the UK Biobank, again this isn't covered in Zheng et al.

Overall, this is an interesting, well conducted study, that substantially advances the literature.

Minor comments

The SNP selection is non-standard in two ways. First they're using genetic variants that are not genome-wide significant. This is not necessarily a problem, but they will need to

report F-statistics to demonstrate that the SNPs are sufficiently predictive of the exposure. Second they use a very liberal threshold for LD ($R < 0.8$), the default in the literature is $R^2 < 0.001$. The authors should clarify whether they're using R , not R^2 , and also why they're using such a low threshold, and that they use methods that allow for correlated variants in the results section (this is well covered in the methods, but (thanks mainly to Nature's curious IRDM format) your readers only discover this right at the end of your paper.

Throughout the phrase "Causal association" should be avoided. Generally the literature discourages this form of scientific euphemism. Here we are estimating causal effects, it is clearer to be explicit about this aim. The MR estimates are not associations, they're ratios of two associations, and are estimates of the causal effect of sclerostin levels on the outcomes. Hernan is very good on this here:

<https://www.ncbi.nlm.nih.gov/pmc/articles/PMC5888052/>

3. Line 138 "For categorical outcomes" - none of the outcomes here appear to be categorical, you have survival and binary outcomes.

We thank the Reviewers for their revision and comments provided.

Pages and lines throughout the response refer to the manuscript without tracked changes.

Reviewer #1 (Remarks to the Author):

This paper reports findings from an MR study suggesting that a reduction in sclerostin level leads to an increased risk of MI and a number of CVD-related risk factors. This finding is potentially important as it adds to other evidence pointing to an increased CVD risk following treatment with sclerostin inhibitors, of which romosozumab is currently being marketed to treat osteoporosis. A similar study has recently been published (J Zheng et al, Arthritis and Rheumatology 2023 doi: 10.1002/art.42538), based on a slightly smaller sample size for the sclerostin GWAS providing the genetic instrument for circulating sclerostin. Zheng et al found similar effects on MI and T2DM risk, but null effects for other outcomes reported here. Specific points are as follows:

Line 69: Having obtained the largest dataset to date of genetically predicted sclerostin levels, the authors focused on cis SNPs as genetic instruments, which makes sense as a strategy for reducing horizontal pleiotropy but would benefit from some explanation/justification.

Thank you for your comment. We have added the following sentence to justify the use of cis-variants as IV variables for our analysis (page 2, line 73-75):

“Specifically, when the exposure under investigation is a drug target, the selection of instruments is usually restricted to those around the gene encoding the drug target to reduce the risk of horizontal pleiotropy²⁴. MR can therefore be used to triangulate the evidence for a causal relationship between sclerostin inhibition on cardiovascular risk²⁵.”

The introduction could mention previous related MR studies; as well as the Zheng paper (included as a preprint here but important to cite the final version as there were some significant differences), papers by Bovijn et al Sci Transl Med 2020; 12:eaay6570 and G Holdsworth et al J Bone Miner Res 2021; 36: 1326–39 could also be cited.

Thank you for your comment and provided references. Aligned with this comment, we have:

- Modified the second paragraph of the introduction (page 2, lines 51-60), to mention two more phase 3 randomised control trials that were conducted.
- Added a paragraph in the introduction, page 2, lines 61-66:
“Some observational studies have also reported similar findings. A meta-analysis of clinical trials and human genetics¹⁹ indicated a potential higher risk of cardiovascular adverse events, and other risk factors such type 2 diabetes. These associations were also supported by a recent Mendelian randomisation study²⁰, where it was found that lower sclerostin levels were causally related with increased coronary artery disease and myocardial infarction risk. However, no evidence was found of these associations in another population-based phenome-wide association (PheWAS) study²¹.”

Line 92 - a p values threshold of 10⁻⁶ was applied for selecting cis SNPs, which requires further justification (the convention is to apply 5 x 10⁻⁸)

Thank you for the comment. We have added further justification in page 3, lines 100-102:

“As we focused on the selection of variants to a single region of the genome, a p-value threshold of 1·10⁻⁶ was prespecified²⁶, which is distinct from the typical GWAS significance threshold: 5·10⁻⁸.”

Line 92 - "r" is used throughout instead of "r²"

Thank you for highlighting this. We have updated this accordingly.

Line 92 - r² of <0.8 was used for pruning. Use of a relatively high threshold has been suggested in these types of analyses in order to boost the number of instruments. However, unless steps are taken to adjust the IVW analysis for correlations between instruments, this approach underestimates the causal standard error. I suspect that many of the positive findings they report are due to relatively narrow confidence limits as they have failed to take account of correlation between instruments.

Thank you for your insightful suggestion. We agree and have updated the method as suggested.

Specifically, we reduced the r² threshold from 0.8 to 0.3 following the literature practical recommendation (Stephen Burgess about *cis*-Mendelian randomisation):

<https://www.mendelianrandomization.com/index.php/blog/69-mendelian-randomization-with-highly-correlated-genetic-variants-cis-mr>

where it states: *"From experience, pruning at r² < 0.3 is generally safe, and r² < 0.4 is usually okay – but I've seen problems at this level in a couple of examples. "*

In this revised version, we additionally performed several sensitivity analyses with r² ranging from 0.001 to 0.8. The results are robust and consistent with our main findings. Please, see page 4, lines 150-162, and page 11, lines 407-412.

The structure of the manuscript is somewhat confusing given the second section refers to additional MR studies using UK Biobank data, whereas the first section also uses UK biobank data. Eg the heel BMD data in Table 2 is exclusively UK Biobank data, with similar analyses based on a subset of the data repeated in Table 3.

Thank you for your comment. We have re-structured the reporting of comprehensive results in the updated manuscript:

- (1) Positive control outcomes: Including heel bone mineral density (source of data: GWAS from previous literature), and hip fracture risk (source of data: GWAS from previous literature).
- (2) Study outcomes based on literature summary statistics: including coronary artery disease, myocardial infarction, ischaemic stroke, hypertension, type 2 diabetes mellitus, LDL cholesterol HDL cholesterol, fasting glucose, and HbA1c. Notice that LDL, HDL, fasting glucose, and HbA1c GWAS have been incorporated in this new version of the manuscript.
- (3) Study outcomes based on UK Biobank data: including cholesterol, LDL and HDL cholesterol, apolipoproteins A and B, lipoprotein (a), C-reactive protein, triglycerides, glucose, HbA1c, coronary artery disease, myocardial infarction, ischaemic stroke, hypertension, and type 2 diabetes.

Also, following the reviewer's suggestion, we have removed the analysis of UK Biobank data for the "heel bone mineral density" and "fracture risk" outcomes, because it simply overlaps with literature summary statistics.

Line 233 and following - see above - findings from Zheng et al need to refer to the published paper

rather than the preprint. Overlap of samples used in these meta-analyses with Zheng et al could also be mentioned (Interval and Fenland were used in both).

Thank you for your comment. We have changed the reference from Zheng et al. and mentioned the overlap of samples used in the meta-analysis (page 6, lines 227-239):

“A previous study²⁰, which also performed a cis-MR approach using a meta-analysis of sclerostin levels in Fenland and Interval studies, reported comparable findings regarding the biomarkers. Despite the wide confidence interval (including the null), their direction of effect suggests increased triglycerides and apolipoprotein B, and reduced levels of protective HDL and apolipoprotein A. Similarly, they found a higher risk of type 2 diabetes mellitus among UK Biobank participants with lower predicted levels of sclerostin. However, Zheng et al. did not find evidence of an association with coronary artery disease, whereas we observed an increase on its risk. Their results also suggested an increased risk with myocardial infarction, whereas in our study this association is borderline significant. A previous meta-analysis of genetic studies found similar associations¹⁹.“

Line 270: I didn't quite understand this point - if this is about simulating different effects of sclerostin inhibition, this refers to a scaling function which doesn't affect statistical power.

Thank you for highlighting this, we have removed it. Additionally, we have added another potential limitation in this study that we failed to consider in the previous version. Please refer to page 7, lines 252-267 for further details.

Reviewer #2 (Remarks to the Author):

Alcalde-Herraiz et al conducted a comprehensive analysis to estimate the effect of sclerostin inhibition on cardiometabolic diseases and risk factors using both Mendelian randomization (MR) and observational analysis. The findings suggested that sclerostin inhibition increased bone health but may increase T2D/glucose, lipids levels, and increase risk of CAD and MI. This aligns with some recent findings using similar pQTL MR setting (but slightly different exposure and some more outcomes). There are some sections that can be improved to reach the standard of Nature Communications.

Major comments:

Please provide a completed STROBE-MR checklist. The authors say they submitted it at the end of the methods section. It should be sent to the reviewers.

Thank you for the comment. We have completed the STROBE-MR checklist together with the revisions.

Meta-analysis of sclerostin GWAS in the cis region. Is there any standard QC been conducted before the meta-analysis? E.g. using some tools to standardize the alleles to a reference panel. Fixed effect meta-analysis results were reported and used for the MR. However, some SNPs showed evidence of heterogeneity (e.g. $I^2=0.74$ for rs66838809). What is the SNP effects of these SNPs with heterogeneity in each cohort? There is evidence to show that SOMA logic v3 and v4 have very different effect estimates. Have this been considered here? Or any other reasons why there are heterogeneity across studies?

Thank you very much for this comment.

We have implemented an exhaustive QC before the meta-analysis was conducted. This QC is reported in page 9, lines 332-337:

"We conducted a meta-analysis of three GWAS of circulating sclerostin levels. Only SNPs within chromosome 17 were included in the meta-analysis, as this is the chromosome where the SOST gene is located. Multiallelic SNPs, variants with invalid IDs, with the same allele for "effect" and "other", with alleles "D" or "I", with alleles frequency ≤ 0 or ≥ 1 , P values < 0 or > 1 , and with standard errors ≤ 0 or equal to infinity, were removed. Additionally, all alleles were standardised to a reference allele using software GWAMA⁴⁸ and METAL⁴⁹."

We have detailed instruments effects in each cohort in Supplementary Table 2.

Regarding the observed heterogeneity to some extent for a few SNPs studied across cohorts, we have evaluated it in two distinct ways:

- We have calculated the genomic inflation factor for the fixed-effects meta-analysed results. The genomic inflation factor was used to assess if there were unmeasured biases in source GWAS study. The result was $\lambda=1.04$, suggesting little residual population stratification. We have reported this in the manuscript:
 - o Page 3, lines 92-94: *"We found little evidence of genomic inflation ($\lambda=1.04$) in the meta-analysed fixed-effects dataset, suggesting that there was little residual genetic bias between the meta-analysed datasets (Supplementary Figure 2)."*

- Page 9, 340-343: “We also used I^2 , Cochran’s Q statistics and Cochran’s Q P value to assess heterogeneity. We calculated the genomic inflation factor to assess whether there was genetic bias within our fixed-effects meta-analysed dataset.”
 - Supplementary Figure 2 shows the QQ-plot.
- We have also conducted a sensitivity analysis using the random-effects method to select instruments. Results were aligned with the main analysis based on fix-effects model. This has been included in the methods section (page 11, lines 413-418) and in the results section: (page 5, lines 163-172).
 - Furthermore, to address more exhaustively the issue regarding heterogeneity, we have conducted an MR analysis using the instrument-sclerostin estimates of each sclerostin GWAS, separately. We have only used GWAS summary statistics for the outcomes. Estimated causal effects are shown in the tables below. All results were aligned with the ones obtained in the main analysis.

Outcome	Beta			SE		
	Ferkinstad	Pietzner	Sun	Ferkinstad	Pietzner	Sun
Bone mineral density	1.33	0.60	0.93	0.05	0.02	0.04
HbA1c	0.03	0.01	0.02	0.04	0.02	0.03
Glucose	0.03	0.02	0.06	0.05	0.02	0.04
LDL	0.03	0.01	0.01	0.04	0.02	0.03
HDL cholesterol	-0.20	-0.08	-0.10	0.04	0.02	0.03

Outcome	OR			SE		
	Ferkinstad	Pietzner	Sun	Ferkinstad	Pietzner	Sun
Hip fracture	0.09	0.33	0.15	0.44	0.20	0.36
Coronary artery disease	1.36	1.14	1.16	0.15	0.07	0.11
Myocardial infarction	1.48	1.21	1.41	0.22	0.10	0.18
Ischaemic stroke	1.29	1.09	1.01	0.21	0.09	0.17
Hypertension	1.03	1.02	1.04	0.03	0.01	0.02

Type 2 diabetes	1.67	1.24	1.28	0.19	0.08	0.14
-----------------	------	------	------	------	------	------

The MR analysis using UK Biobank data. It is interesting to try to use survival analysis results as outcome. I think there are some GWAS methods, e.g. GATE that can be used, which avoiding some potential biases.

We sincerely appreciate your suggestion. The GATE survival method may primarily be advantageous in scenarios with substantial censoring data. However, we argue that this is not a case in our study based on the UK Biobank cohort, in which only a few participants were withdrawn during the follow-up period.

Aligned with the reviewer’s suggestion, we explored alternative packages for conducting an analysis similar to the one we intended. Our research led us to the *gwasurvivr* package from Bioconductor platform:

<https://academic.oup.com/bioinformatics/article/35/11/1968/5161085?login=true>.

Upon careful examination of the methodology employed in this package, we observed a similarity to the approach we have been using. This fact has strengthened our confidence in the robustness of our method, as results obtained with the package match the ones obtained with our analysis. Please, refer to the attached figure that compares the MR estimates obtained using our method and the mentioned package:

For the UK Biobank outcomes, it is unclear why we have to use UKBB outcomes? E.g. for lipids traits, we can use GLGC data, for glucose traits, we can use MAGIC data, which are measured after fasting. All these data have larger sample size, which may increase reliability and power.

Thank the reviewer for this suggestion. We have incorporated these GWAS into the analysis, please see results section (page 3, lines 120-121), methods section (page 10, lines 369-374), figure 3, table 2, and table 3.

The generalised IVW method. The confidence interval of the MR estimates seems to be narrow than other studies. I wonder if even small differences in LD between the discovery sample and the outcome sample in a two sample MR could lead to some double counting, if the signals are all coming from LD with a single non-measured causal variant.

Thank you for highlighting this. The confidence interval of the MR estimates was narrow than other studies because of an error we made in the statistical code. We have now corrected this and updated all the results accordingly.

Except the generalised IVW, some other methods that better control for weak instrument bias can be considered, e.g. cML-MA, debiased IVW and MRAID

Thank you sincerely for the suggestion. We argue that the weak instrument bias is unlikely an issue in our study. First, we have calculated the F-Statistic for each one of the instruments, and for all the two instruments (rs7220711, rs66838809) used for the meta-analysis, F-Statistic > 10. Also, both instruments had a strong association with both positive control outcomes.

Additionally, we have implemented a sensitivity analysis using the Principal Components approach, which uses the first principal components that explain more than 99.9% of the genetic variance observed to perform the Mendelian randomisation analysis. This approach is less susceptible to weak instrument bias and produce similar results. Please refer to the literature for more details: <https://onlinelibrary.wiley.com/doi/10.1002/gepi.22506#gepi22506-bib-0019>

The survival analysis, it is an interesting idea to do a survival analysis, which will be helpful for evidence triangulation. However, this section needs more explanations. For example, what is the exposure of this survival analysis? Usage of sclerostin inhibitors? Circulating sclerostin levels or PGS of sclerostin inhibition? What is the control group of this analysis? Non-users? What is the average follow-up period? For covariates, why only sex and first 10 PCs were used? Please consider other potential confounding factors that may bias the results, e.g. study centre, age, usage of other drugs etc.

Thank you for your comment. The exposure in our study remains the circulating sclerostin level. The primary difference in survival analysis is that we used the Cox model to calculate the hazard ratios between genetic variations and outcomes of interest, as opposed to the odds ratios derived from logistic regression. The average follow-up period for each disease outcome is provided in Supplementary Table 6.

In the original version, we adjusted for sex, and the 10 principal genetic components to account for population stratification in all genetic analysis models. Following the reviewer suggestion, we have also added the age at first assessment and the genetic batch into the model. The new results have been updated throughout the manuscript yet showing little change after this addition.

Colocalization, weak evidence was observed, which could be a power issue for the outcome data. or maybe too many cis signals in the same region? Will a suise+coloc pipeline be helpful here?

Thank you very much for your suggestion. Reviewing literature, we are hesitant to use SuSiE, as it is very sensitive to the choice of LD matrix used. Further, *coloc* package has been proven to be fairly

robust to violations of the single variant assumption
(<https://journals.plos.org/plosgenetics/article?id=10.1371/journal.pgen.1004383>).

Based on our research experience and literature
(<https://www.sciencedirect.com/science/article/pii/S0002929722001495?via%3Dihub#sec9>), we suspect that in most of the analyses the issue relies on the lack of power to detect colocalization.

Minor comments:

Title is unclear, “cause-effect relationship” is vague. “UK Biobank analysis” is an observational analysis?

We have changed the title to:

“Effects of sclerostin levels on cardiovascular biomarkers, risk factors and cardiovascular disease: a summary-statistics based Mendelian randomisation study”.

Abstract, causal association. Not a nice combination given the conceptual difference between causality and association.

We have changed this concept to: “*causal relationship*”. We have also changed it throughout all the manuscript.

Introduction, need to cite MR studies of sclerostin and explain the novelty of the current study clearly.

Thank you for your comment. We have added the following sentence (page 2, lines 63-66), where we cite the previous MR study of sclerostin that has been conducted:

“These associations were also supported by a recent Mendelian randomisation study²⁰, where it was found that lower sclerostin levels were causally related with increased coronary artery disease and myocardial infarction risk. However, no evidence was found of these associations in another population-based phenome-wide association (PheWAS) study²¹.”

We have also explained the novelty of the current study in page 2, lines 80-85:

“The novelty of this study relies on the use of the most up to date sclerostin GWAS (at the time of the study) for the IVs selection, and the inclusion of UK Biobank data for evidence triangulation. We also study effects on biomarkers, cardiovascular traits, and risk factors to provide insights about the underlying mechanistic insights. Moreover, we performed colocalization analyses to study the probability of a shared causal variant between the investigated traits.”

Reviewer #3 (Remarks to the Author):

This is a nice application of MR that investigates the relationship between sclerostin and a range of disease outcomes. This is a nice addition to the literature that goes considerably beyond other recent papers. While superficially similar, this paper is a considerable advance on Zheng et al, which is reflected in the definitive and precise results provided. For example, the current study estimates an odds-ratio for MI of 1.28 (95%CI: 1.18, 1.38) versus 1.35 (95%CI: 1.01-1.79) in Zheng et al. I.e. the current study has standard errors that are 3.6 fold smaller on the log-odds scale. This is roughly the equivalent to a 13 fold increase in sample size. It would be helpful for your readers to be explicit about where the additional power from your study comes from (the sample size for your discovery GWAS is slightly larger, so this is unlikely to explain it).

The study also presents additional novel data from the UK Biobank, again this isn't covered in Zheng et al.

Overall, this is an interesting, well conducted study, that substantially advances the literature.

Minor comments

The SNP selection is non-standard in two ways. First, they're using genetic variants that are not genome-wide significant. This is not necessarily a problem, but they will need to report F-statistics to demonstrate that the SNPs are sufficiently predictive of the exposure.

Thank you for your comment. We have reported the F-statistics in Table 1.

Second they use a very liberal threshold for LD ($R < 0.8$), the default in the literature is $R^2 < 0.001$. The authors should clarify whether they're using R , not R^2 , and also why they're using such a low threshold, and that they use methods that allow for correlated variants in the results section (this is well covered in the methods, but (thanks mainly to Nature's curious IRDM format) your readers only discover this right at the end of your paper).

Thank you for highlighting this. We have now stated clearly that we are using an r^2 , not r .

Beforehand, we would like to emphasize that we have modified r^2 from 0.8 to 0.3, as explained earlier in this document from reviewer 1. Furthermore, in page 3, line 103, we have now explicitly explained our objective of identifying low-correlated variance, hence justifying the utilization of this threshold:

"We used pruning to select low-correlated variants ($r^2 \leq 0.3$)".

Throughout the phrase "Causal association" should be avoided. Generally the literature discourages this form of scientific euphemism. Here we are estimating causal effects, it is clearer to be explicit about this aim. The MR estimates are not associations, they're ratios of two associations, and are estimates of the causal effect of sclerostin levels on the outcomes. Hernan is very good on this here: <https://www.ncbi.nlm.nih.gov/pmc/articles/PMC5888052/>

Thank you for reporting this and the provided reference. We have changed the concept "causal association" to "causal relationship" throughout the manuscript.

Line 138 "For categorical outcomes" - none of the outcomes here appear to be categorical, you have survival and binary outcomes.

Thank you for highlighting this. We have changed the description of the outcomes accordingly.

Reviewers' Comments:

Reviewer #1:

Remarks to the Author:

In this revised version, the authors have corrected an error in their analysis code which had led to erroneous confidence intervals for some of their IVW analyses. This has resulted in attenuation of several of the causal effects of sclerostin lowering observed previously, including risk of coronary artery disease and MI, which are now either close to unity or cross it. This inevitably reduces the overall impact of the paper. On the other hand, other changes in response to the reviewer comments have generally improved the paper.

Reviewer #2:

Remarks to the Author:

The authors have answered most of my questions with literature search and try to find the best solution/method/model for each analysis. I think the manuscript is generally in a better shape now. I have several additional comments here:

1. better to comment on the recent pre-print paper on this topic

<https://www.medrxiv.org/content/10.1101/2024.01.30.24302021v2>

2. the MR analysis of each pQTL study separately showed that DeCODE data generally showed larger odds ratio compared to other two studies. Are all data been standardised into the same unit or using similar covariates in the GWAS model of sclerostin? Also, OR for fracture in DeCODE was 0.09? Is this way too low than expected?

3. coloc analysis, I think this is important since there are multiple pQTL signals in the cis SOST region. There are few new methods coming out, including SharePro, which may help. Also, need to comment on why MR found robust evidence but coloc not, which power is just one issue. MR is able to combine effect of conditional independent signals together using a meta-analysis model, which is very helpful to increase power in this case. To me, a new coloc model is needed that can estimate combined causal effects, which Steven Bruggess's method proportional coloc may talk to this?

Reviewer #3:

Remarks to the Author:

Thank you for addressing my comments.

We thank the Reviewers for their revision and comments provided.

Reviewer #1 (Remarks to the Author):

In this revised version, the authors have corrected an error in their analysis code which had led to erroneous confidence intervals for some of their IVW analyses. This has resulted in attenuation of several of the causal effects of sclerostin lowering observed previously, including risk of coronary artery disease and MI, which are now either close to unity or cross it. This inevitably reduces the overall impact of the paper. On the other hand, other changes in response to the reviewer comments have generally improved the paper.

Thank you very much for the positive comment.

Reviewer #2 (Remarks to the Author):

The authors have answered most of my questions with literature search and try to find the best solution/method/model for each analysis. I think the manuscript is generally in a better shape now. I have several additional comments here:

1. Better to comment on the recent pre-print paper on this topic:

<https://www.medrxiv.org/content/10.1101/2024.01.30.24302021v2>

Thank you very much for the literature suggestion.

This recent pre-print studied the impact of circulating sclerostin levels on heel bone mineral density, osteoporosis, and multiple cardiovascular outcomes based only on UK Biobank data.

Our study differs from theirs in many aspects. For example, we incorporated multiple GWASs, instead of only one in the pre-print, to identify reliable instrumental SNPs for circulating levels of sclerostin, which is the foundation of robustness and generalisability of findings from MR study. As a result, one of the IVs variables (rs6416905) in their study did not reach statistical significance in ours after meta-analysis (P-Value= $8 \cdot 10^{-6}$) and could be a major source of weak IV bias.

Also, the one-sample Mendelian randomisation method utilised in their study is commonly known to be susceptible to overfitting and bias.

2. the MR analysis of each pQTL study separately showed that DeCODE data generally showed larger odds ratio compared to other two studies. Are all data been standardised into the same unit or using similar covariates in the GWAS model of sclerostin? Also, OR for fracture in DeCODE was 0.09? Is this way too low than expected?

Thank you very much for your comment.

All the studies used the same units and similar covariates in the GWAS model of sclerostin. Nonetheless, to address the reviewer concern, we have repeated our study but only combing GWAS from the INTERVAL and the Fenland study (excluding of DeCODE data). The results are consistent with the main analysis, showing the same directional trends and statistical significance, albeit slight variations in values.

*In black: results from the original manuscript. In blue: results excluding DeCODE study to estimate the SNP-Circulating sclerostin effect size.

	Beta		Se		P-Val	
Heel bone mineral density	1.00	0.66	0.04	0.02	9e-137	7.6e-139
Hip fracture	0.16	0.287	0.33	0.22	2e -08	1e-8
Coronary artery disease	1.25	1.15	0.11	0.07	0.04	0.06
Myocardial infarction	1.35	1.24	0.17	0.11	0.07	0.05
Ischaemic stroke	1.18	1.09	0.16	0.1	0.26	0.4
Hypertension	1.02	1.02	0.02	0.01	0.18	0.1
Type 2 diabetes	1.45	1.25	0.14	0.09	0.01	0.01

3. coloc analysis, I think this is important since there are multiple pQTL signals in the cis SOST region. There are few new methods coming out, including SharePro, which may help. Also, need to comment on why MR found robust evidence but coloc not, which power is just one issue. MR is able to combine effect of conditional independent signals together using a meta-analysis model, which is very helpful to increase power in this case. To me, a new coloc model is needed that can estimate combined causal effects, which Steven Bruggess's method proportional coloc may talk to this?

Thank you very much for you comment. We believe that use of Colocalisation (specifically fully Bayesian colocalization) nowadays is common practice in supporting Mendelian randomisation findings.

Although SharePro is a novel method that uses an effect group-level approach to infer posterior probabilities of colocalization, it has not been validated by wider research community. Hence, we feel less confident to apply this approach in our study.

Instead, we have performed a sensitivity analysis for the potential power issues in colocalization. Specifically, we analyse how the prior probability of a shared causal variant (p_{12}) impacts the posterior probabilities. Our hypothesis was defined as:

- H0: No association with none of the traits.
- H1: Association with trait 1 (i.e., circulating sclerostin).
- H2: Association with trait 2 (i.e., outcome).
- H3: Association with both traits but at separate causal variants.
- H4: Association with both traits at a shared causal variant.

In subsequent plots, the green area corresponds to $H4 > H3$.

In our sensitivity analysis, we observed strong evidence of colocalization with the positive control outcomes, even with small prior probabilities of a shared casual variant:

Bone mineral density

Hip fracture

Although colocalization signals for the outcomes where Mendelian randomisation pointed to a causal relationship were not strong, we saw that when increasing the value of the prior probability p_{12} , there was an increase in the posterior probability of a shared causal variant.

Coronary artery disease

Type 2 diabetes

This behaviour was not seen when analysing the HDL-Cholesterol colocalization results, where we found strong evidence of H3 (Association with both traits but at separate causal variants):

HDL

Based on these results, we would expect that low power rather than genetic colocalization could be the major reason for the inconsistency between the MR with coloc findings.

Reviewer #3 (Remarks to the Author):

Thank you for addressing my comments.

Thank you very much.

Reviewers' Comments:

Reviewer #2:

Remarks to the Author:

Thank you for the authors to address my comments. The attempts to explain the weak colocalization evidence is caused by low power is helpful, which can be set as a sensitivity analysis for future colocalization analysis.

Just one small error can be corrected, the green area was not correctly drawn for T2D?
The overlap point between H3 and H4 is around $1e-5$.

We thank the Reviewers for their revision and comments provided.

Reviewer #2 (Remarks to the Author):

Thank you for the authors to address my comments. The attempts to explain the weak colocalization evidence is caused by low power is helpful, which can be set as a sensitivity analysis for future colocalization analysis.

Just one small error can be corrected, the green area was not correctly drawn for T2D? The overlap point between H3 and H4 is around $1e-5$.

Thank you very much for your comment. The green area corresponds to $P(H4) > P(H3)$, so the interpretation of the following figure points that with higher values of prior shared probability (i.e., around $5 \cdot 10^{-5}$) we would have had $P(H4) > P(H3)$. However, with smaller values (i.e., $1 \cdot 10^{-5}$), we do not observe that.